# Attitudes Toward In-Training Assistance Dogs in University Classrooms

**DOI:** 10.3390/ani15233476

**Published:** 2025-12-02

**Authors:** Lindsey Person, Greg C. Elvers

**Affiliations:** Department of Psychology, College of Arts and Sciences, University of Dayton, Dayton, OH 45469, USA; lindseyperson@gmail.com

**Keywords:** puppy raisers, in-training assistance dogs, dogs in the classroom

## Abstract

Assistance dogs provide many important benefits and improve the quality of life of the humans they serve. Before they can be an assistance dog, dogs must undergo training. An initial part of the training involves learning basic commands (e.g., sit) and exposing the dog to many different people and situations. This initial training is sometimes performed by university students. When university students bring dogs to classes, there may be problems; for example, other students may be allergic to the dog, fearful of the dog, or have religious objections to the dog. Students who bring dogs to class may also experience problems such as unwanted attention to the dog, disruptions to learning due to needing to attend to the dog, and missing class due to the dog needing veterinarian care. The dogs may also experience stress from being in novel situations with novel people. This study surveyed instructors, students, and the primary caretakers of the dogs in classes visited by the in-training dogs. In general, students reported having positive attitudes about having dogs in class; the students responsible for the dog reported few problems and that their dogs experience few, if any, signs of stress. The results suggest that university students can continue to provide the initial training for dogs without negative consequences to the other students in the classes visited by the dogs.

## 1. Introduction

Assistance dogs, also known as service dogs, provide valuable services to many groups. People who are blind or have impaired visual acuity use guide dogs to help them navigate their environment [1,2]. Hearing dogs alert people who are deaf or have hearing impairments to key sounds such as a doorbell or a ringing telephone [3,4]. Mobility assistance dogs help people who have limited or no use of their extremities or issues with balance to retrieve items, move up or down ramps, open doors, and operate light switches [5,6]. Medical response dogs have been trained to detect hypoglycemia, hyperglycemia, seizures, migraines, asthma, and narcolepsy and to alert the person often before the person is aware of the onset of the condition [7,8]. Autism assistance dogs help people who have autism spectrum disorder to remain safe and to live more independently [9,10,11]. In addition to the specific service that assistance dogs provide, they also improve the psychological wellbeing, emotional functioning, and self-esteem of their owners [12].

Before dogs can be certified as assistance dogs, they undergo two types of training: general training (socialization and desensitization; obedience training) and specialized training for the group that they will eventually serve. The general training often occurs when a dog is still a puppy and is an important first step toward becoming an assistance dog. The goal of socialization and desensitization is to familiarize the in-training assistance dogs with as many types of people, situations, and environments as possible so the dogs will remain calm when they are around people they have never seen or in new situations or environments. Obedience training teaches the dogs the basic commands (e.g., sit, stay, come, down, and leave it) so that the dog can follow instructions and stay on task.

Organizations that train assistance dogs often have community members, called puppy raisers in the literature, who assist with the general training that the dogs receive during their first year of training. Many groups act as puppy raisers: individuals in the community [13,14], prisoners [15,16], members of organizations/clubs (such as high school students in Future Farmers of America) [17], and university students [18,19]. Anecdotally, some university students volunteer to raise puppies because they love dogs and often have a strong commitment to help others. Puppy raisers go to many daily activities and take the in-training assistance dogs with them to expose the dogs to as many people and situations as possible. For university students who are puppy raisers, this includes taking the dogs to classes.

Bringing in-training assistance dogs to class may or may not be problematic. Given the number of households that own dogs, most people in the United States, including instructors and students, like dogs [20] and might believe that having a dog in class is not only acceptable but is enjoyable. However, bringing in-training assistance dogs to class may be problematic for some students and instructors of classes visited by the dogs. In a questionnaire study of adults living in Adelaide, Australia, almost half of a sample of 292 adults reported some fear of dogs [21]. This fear may hinder students’ ability to pay attention in class or the instructor’s ability to teach. Usually, such fears are relatively mild [21] but possibly could be heightened by an in-training assistance dog that has not fully learned the basic obedience commands.

Morris reports that the prevalence of allergies to pet dander, either cat or dog, increased rapidly from 1950 to 2010 [22]. It is estimated that 10 to 20% of people are allergic to dogs and/or cats [23]. In a classroom of 30 students, it is possible that multiple students will have allergies to dogs. Because the dander will likely stay in the classroom until the room is cleaned, bringing a dog to class could cause an allergic response in many students who use the same classroom later in the day. Common allergic reactions such as sneezing and watery eyes might impair such students’ ability to fully participate in class.

While rare overall, and rarer still in dogs that receive regular veterinarian care such as puppies being raised to possibly become assistance dogs, zoonotic diseases can be transmitted from dogs to humans. Zoonotic diseases that can be spread from dogs to humans through contact with urine or feces include Leptospirosis, noroviruses, and Salmonella [24,25]. Touching the skin or fur of an infected dog can spread ringworm and dermatophytosis [24,25]. Lyme disease can be spread through tick or flea bites and dog bites can spread rabies and Pasteurella [24,25]. Thus, students in classes that are visited by dogs have some risk, albeit likely low, of contracting a zoonotic disease. Because many people are unaware that dogs can transmit a variety of diseases to humans [26], some people may not take appropriate safety measures, such as washing hands, after interacting with a dog in the classroom. Some zoonotic diseases may pose extra risks for immunocompromised students in classrooms that in-training assistance dogs visit.

The European Pet Food Industry Federation estimates that 25% of European households own at least one dog [27]. Given the popularity of dogs as pets, it would not be surprising if many students and instructors in a class visited by an in-training assistance dog would want to look at and/or interact with the dog. That is, some students who like dogs might sometimes find in-training assistance dogs in the classroom distracting and might pay attention to the dog rather than paying attention to classroom activities.

Berglund states that some Muslims consider dogs to be unclean and thus Muslims sometime have negative attitudes toward dogs [28]. While some Muslims might be offended by having a dog in a classroom, Berglund states that working dogs (dogs used for guarding or hunting) and guide dogs are generally accepted by Muslims [28]. The negative attitude toward dogs, in general, might be shifting positively as a result of Western influences such as television and movies in Muslim areas [28].

Because some of the younger in-training assistance dogs will not be fully trained to sit and be quiet for the duration of a class (over an hour in some cases), they might cause disruptions in class by failing to hold the down command, rattling tags on their collar, playing with toys, or making noises. Such distractions might make it harder for instructors and students to stay focused on classroom activities.

Assistance dogs are often guaranteed certain rights by law in some countries such as the Americans with Disabilities Act in the United States [29,30]. The rights often allow assistance dogs to go into places where other dogs are typically forbidden such as restaurants and hospitals. At least in the United States, in-training assistance dogs typically do not enjoy these rights by federal law but many states have laws granting such rights [29,30]. Some legal scholars question whether dogs undergoing the general training are considered in-training assistance dogs because they have yet to start the training for the specific duties that they will perform as assistance animals. If they are not legally considered in-training assistance animals, they may not have the legal rights to be in the classroom [30]. If a dog in general training is neither an assistance dog nor an in-training assistance dog, allowing them in classrooms may expose universities to legal liability if the dog injures a person.

While the above issues may make some students and instructors uncomfortable with having in-training assistance dogs in the classroom, the severity of the discomfort and the number of students and instructors affected must be weighed against the magnitude of the benefits provided by the in-training assistance dogs now and in the future. While the in-training assistance dogs cannot yet provide specific services like an assistance dog, they can still provide valuable services to instructors and students in a classroom. For primary (ages 5 to ~11 years) and secondary (ages 11 to ~18 years) students, having a dog in the classroom leads to positive learning outcomes [31], enhanced social and emotional development, improved attitudes toward academia [32], and reduced stress level as measured by salivary cortisol [33]. While the preceding studies look at younger students, it may be that these benefits also apply to older students. At the university level, interacting with dogs in class decreased anxiety, improved mood scores [34] and increased participation in class [35].

Not all in-training assistance dogs will eventually become assistance dogs, but for the subset that do, they will provide many important benefits to the person they serve [36]. For example, assistance dogs for children on the autism spectrum improve the child’s self-regulation and participation in daily routines [37]. Assistance dogs for children on the autism spectrum also improve the lives of the child’s family and reduce stress in the parent of the child [38,39]. Hearing dogs improve the ability of people who are deaf or have profound hearing loss to perform daily tasks and respond appropriately to dangerous situations (e.g., a smoke alarm going off) [40]. Assistance dogs can reduce stress for caregivers of people with mobility issues [41]. Besides providing specific services such as guiding a person who is blind, assistance dogs can also have positive impacts on physical and psychological wellbeing, quality of life, decreased stress, and social inclusion [12,42,43,44]. It is reasonable to conclude that the quality of life would be significantly worse if assistance dogs did not exist for those who need them and their families.

While it may be difficult to compare the costs to the benefits of having in-training assistance dogs in the classroom, there may be simple ways of minimizing some of the costs which in turn could tip the scale in favor of having such dogs in class. Simple activities could be implemented to reduce the costs such as announcing the days when the in-training assistance dogs will be in class so students who are allergic to dogs could take medicine to reduce their allergic reactions. The puppy raiser and in-training assistance dog could sit near the rear of the class so that fewer students would see the dog unless they turned around. The puppy raiser could affix tags (identification, rabies vaccination) on the in-training assistance dog’s collar so that they do not bang into each other making noise.

The puppy raiser of the in-training assistance dog also may have potential issues with bringing the dogs to class. If the puppy raiser must stop paying attention to class and manage the in-training assistance dog’s behavior, the puppy raiser might miss important information that is being presented by the instructor of the class. It might only take a few seconds to issue a new command to the in-training assistance dog or it could possibly take several minutes to remove the dog from the classroom until the dog can behave again. The puppy raiser may receive unwanted attention from students and instructors who want to interact with the in-training assistance dog. If the in-training assistance dog becomes ill, the puppy raiser might have to miss class to take care of the dog and/or take the dog to the veterinarian. Some of these issues may be at least partially mitigated by leaving the in-training assistance dog with another puppy raiser when the need arises.

There may be negative consequences for the in-training assistance dog itself. Like humans, dogs have distinct personalities—relatively enduring differences in behavior across individuals [45]. While organizations that train assistance dogs usually breed dogs with the expectation that they will be calm and obedient, it is not always possible to predict a dog’s personality based on its genetic heritage. If a puppy is prone to being fearful, being placed in new situations and situations with lots of people who want to interact with the puppy may cause stress, especially early in the puppies’ training before the puppies have been fully socialized and desensitized to novel people, situations, and environments.

Given the limited amount of research on the above topics, this paper explores the following hypotheses:

**H1.** 
*In general, university students will have positive attitudes toward having in-training assistance dogs in class.*


**H2.** 
*University students who have allergies to dogs, are anxious around dogs, or who, for religious reasons, believe that dogs are unclean, will have less positive attitudes toward having in-training assistance dogs in class.*


**H3.** 
*In general, instructors will have positive attitudes toward having in-training assistance dogs in class.*


**H4.** 
*Instructors who have allergies to dogs, are fearful of dogs, or who, for religious reasons, believe that dogs are unclean, will have less positive attitudes toward having in-training assistance dogs in class.*


**H5.** 
*Puppy raisers will experience some negative issues about taking the in-training assistance dogs to class.*


**H6.** 
*At the point in the semester that data were collected (approximately 9 to 10 weeks since the start of the semester), the puppies will be sufficiently socialized and desensitized to show few, if any, signs of stress while in the classroom.*


**H7.** 
*Respondents will believe that there are ways of improving the experience of going to a class in which an in-training assistance dog is present.*


**H8.** 
*Scores on the C-DAS (Coleman Dog Attitude Scale) will be positively correlated with attitudes toward having in-training assistance dogs in class.*


## 2. Materials and Methods

### 2.1. Participants

There were three groups of participants in this study. The first group consisted of 65 students who were taking a class that was frequently visited by an in-training assistance dog. There were 48 females, 13 males, 3 gender variant/non-conforming, and 1 student who preferred to not answer the gender question. For the 51 students who reported their age as between 18 and 21 years, the mean age was 19.8 years with a standard deviation of 1.0 years. An additional 13 students reported that they were 22 years or older and one student preferred not to give their age. One additional student’s data were discarded for failing to appropriately respond to an attention check question about the current year. Eight additional students’ data were discarded as they started, but did not finish the questionnaires.

The second group consisted of nine instructors who were teaching a class that was frequently visited by an in-training assistance dog. One instructor reported their age between 20 and 29 years, two instructors reported their age for each of the categories, 30–39, 40–49, 50–59 years, and 60 years or older. Three instructors reported they were female, five were males, and one preferred not to say.

The third group consisted of five puppy raisers who took their dogs to classes. Their mean age was 20.4 years with a standard deviation of 0.9 years. All were female.

All participants granted consent to participate and were debriefed at the end of the questionnaires. All participants were students or instructors at either a medium-sized, private, religiously affiliated university (n_students_ = 40, n_instructor_ = 7, n_puppy raisers_ = 3) or a medium-sized, public university (n_students_ = 25, n_instructor_ = 2, n_puppy raisers_ = 2) in Ohio. Participants were informed that a single USD 100 donation would be made on behalf of the participants to the organization that provided the in-training assistance dogs to the puppy raisers.

### 2.2. Materials

Students, puppy raisers, and instructors responded to demographic questions about their age (18, 19, 20, 21, 22, 23–29, 30–39, 40–49, 50–59, 60 or more years, prefer not to say), gender (female, male, transgender female, transgender male, gender variant/non-conforming, prefer not to say, other), university attending/working at, religious affiliation (the 10 largest religions [46], other, prefer not to say), dogs that they have had (companion/pet, emotional support, service dog [the term service dog was used throughout the questionnaire in place of assistance dog as service dog is the common term in the sample tested], working dog), and respondent type (student, handler [the term handler was used throughout the questionnaire as this is the common term in the sample for puppy raiser], secondary handler/sitter, or instructor).

Students, puppy raisers, and instructors answered the Coleman Dog Attitude Scale (C-DAS) which measures attitudes toward dogs and consists of 24 questions (e.g., When I see a dog I want to play with it, I avoid dogs) with responses on 5-point Likert scales (1 = strongly disagree, 5 = strongly agree) [47]. Cronbach’s α for the C-DAS ranged from 0.98 to 0.99 across three studies [47]. The convergent and discriminative validity of the C-DAS was established by the findings that the C-DAS predicts whether people intend to interact with dogs but does not predict whether people are willing to donate to dog-related charities or volunteer to help children who are fearful of dogs [47]. These findings suggest that the C-DAS has construct validity. The C-DAS is scored by summing the 24 responses after reverse scaling questions 13 and 24.

Students and instructors answered a 12-item questionnaire asking about attitudes toward having the in-training assistance dog in the classroom. Responses were made on a 5-point Likert scale (1 = strongly disagree, 5 = strongly agree). An open-ended question asked students and instructors to share any comments they had about the in-training assistance dog. Two additional questions were attention check questions to see if respondents were reading the questions before responding. Responses from people who agreed or strongly agreed to either attention check question (the current year is 2028 (data were collected in 2024 and 2025) or that they had three noses) were discarded. The score for positive attitudes toward having in-training assistance dogs in the classroom is the sum of the responses to the 12 questions in Table 1 after reverse scaling the responses to questions 4, 6, 7, 8, 9, and 10. Higher values indicate more positive attitudes toward the dogs in classes.

Students and instructors also answered a seven-item questionnaire (see Table 2) on ways that the experience of having an in-training assistance dog in the classroom might be improved. Responses were made on a 5-point Likert scale (1 = strongly disagree, 5 = strongly agree). An open-ended question asked students and instructors to share any other ideas for improving the experience.

Puppy raisers answered a 14-item questionnaire (see Table 3) that addressed signs of stress [48] in the in-training assistance dog when the dogs were in class. Responses to the first 12 items were made on a 5-point Likert scale (1 = strongly disagree, 5 = strongly agree). The stress score was the sum of the responses to the first 12 questions. Higher scores correspond to more signs of stress in the dogs.

Puppy raisers also answered a 9-item questionnaire (see Table 4) about their experiences with having the in-training assistance dog during class.

### 2.3. Procedure

The in-training assistance dogs were provided to the puppy raisers by 4 Paws for Ability [49], a non-profit organization that trains assistance dogs. 4 Paws for Ability sponsors student organizations on campuses and the presidents of these student organizations were identified via a Facebook group created by 4 Paws for Ability. The student organizations consist of students who are puppy raisers and other students interested in assistance dogs. The presidents of the student organizations were contacted and asked if they were interested in participating in research on attitudes toward having in-training assistance dogs in the classroom. If so, they were asked to provide the course descriptor (e.g., PSYC 100, Section 2) and instructor’s email address for classes that the dogs were taken to on a regular basis. After consulting with the Institutional Review Boards (ethics committees) and gaining permission from the universities, the instructors were emailed a brief description of the research and asked to forward another email with a link to the online questionnaire to the students in the appropriate classes. The instructors were also given a link to the online questionnaire. Unfortunately, due to out-of-date information on the Facebook group, difficulties in getting the student organizations to agree to participate, and issues with gaining approval from other universities to survey their instructors and students, only two universities were included in the data collection. One university is a medium-sized, private, religiously affiliated university, while the other is a medium-sized, public university. Both were located in Ohio, United States, relatively near the location of 4 Paws for Ability.

The questionnaires were administered online with Qualtrics.

### 2.4. Statistical Analysis

Statistical analysis was performed with IBM SPSS version 29. Hypotheses one and three, that university students and instructors will tend to have positive attitudes toward in-training assistance dogs in class were tested with one-tailed, one-sample *t*-tests comparing the attitudes to the neutral response point (3—neither agree nor disagree) of the Likert scale responses to the questions in Table 1. Attitudes were calculated by taking the mean of the 12 questions after reverse scoring the responses to questions 4 and 6 through 10. Even though the responses to the individual questions may not have been normally distributed, once the responses are summed across questions, the measure will be approximately normal by the central limit theorem. Hypotheses two and four, that university students and instructors who are allergic to, anxious of, or who might have negative religious attitudes toward dogs will have less positive attitudes toward in-training assistance dogs than those who do not have these attributes, were tested with one-tailed, two-sample Mann–Whitney *U*-tests. The rank–biserial correlation (*r*_rb_) was used as the effect size. Hypothesis five, that puppy raisers will experience some negative issues concerning taking the in-training assistance dog to class, was analyzed with one-tailed, one-sample *t*-tests comparing the mean of questions 1 through 5 and 8 through 9 in Table 4 after reverse scoring question 8 to the neutral response point (3). Questions 6 (professor interacts with the dog) and 7 (take dog to class on quiz/exam days) were not included as they may not represent problems—professors are busy and have little time to interact with the dogs and puppy raisers typically have a backup person that they can leave the dog with when they are otherwise busy. Wilcoxon signed-rank tests for each individual question were also performed. Hypothesis six, that the in-training assistance dogs will show few signs of stress while in the classroom, was analyzed with one-tailed, one-sample *t*-tests comparing the mean of the first 12 questions in Table 3 to the neutral response point (3—neither agree nor disagree). Wilcoxon signed-rank tests for each individual question were also performed. Hypothesis seven, ways of possibly improving the experiences of university students in classes that are visited by in-training assistance dogs, was analyzed with one-tailed, one-sample *t*-tests comparing the mean of the seven questions in Table 2 to the neutral response point (3—neither agree nor disagree). Wilcoxon signed-rank tests for each individual question were also performed. Hypothesis eight, that scores on the Coleman Dog Attitude Scale will be positively correlated with attitudes toward having the in-training assistance dog in class was tested with Pearson’s correlation.

To help control Type-I errors across the family of inferential tests, α_comparison-wise_ was set to 0.005.

## 3. Results

University students were hypothesized to have positive attitudes toward in-training assistance dogs in class. The mean of the responses to the 12 questions in Table 1, after reverse scoring questions 4 and 6 through 10 was computed for students who completed the questionnaire and who were not puppy raisers; *M* = 4.1, s.d. = 0.6. Larger values indicate more positive attitudes on a 1-to-5 scale. Consistent with the hypothesis, a one-tailed, one-sample *t*-test comparing the means to the value 3 (the neutral point of the Likert responses) revealed a significant difference with a large effect size; *t*(62) = 15.712, *p* < 0.001, Cohen’s *d* = 1.979.

University students who were allergic to, anxious of, or who possibly had religious objections to dogs were hypothesized to have more negative attitudes toward in-training assistance dogs in class than students who did not have any of these characteristics. University students who were not puppy raisers and who completed the questionnaire were separated into two groups. One group consisted of students who agreed or strongly agreed to either question 8 (anxious about dogs) or question 9 (allergies to dogs) in Table 1 or indicated that they were Islamic (who may or may not object to dogs as unclean). The other group consisted of all other students who were not puppy raisers and who completed the questionnaire. Inconsistent with the prediction, a one-tailed, Mann–Whitney *U* -test comparing the attitudes toward in-training assistance dogs in the classroom for the first group (*M* = 3.2, s.d. = 1.1, *n* = 3) to the second group (*M* = 4.2, s.d. = 0.5, *n* = 60) failed to find a statistically significant difference, *U* = 41.500, *p* = 0.062, *r*_rb_ = −0.359. This result should be considered exploratory given the small sample size of the first group.

Instructors were hypothesized to have positive attitudes toward in-training assistance dogs being in class. The mean of the responses to the 12 questions in Table 1, after reverse scoring questions 4 and 6 through 10 was computed for the eight instructors who completed this questionnaire; *M* = 3.2, s.d. = 0.6. Larger values indicate more positive attitudes. Inconsistent with the hypothesis, a one-tailed, one-sample *t*-test comparing the means to the value 3 (the neutral point of the Likert responses) failed to find a statistically significant difference; *t*(7) = 1.101, *p* = 0.154, Cohen’s *d* = 0.389. Changing the hypothesis to a non-directional hypothesis doubles the *p* value and failed to reveal a difference from the neutral point of the scale (3).

Instructors who were allergic to, anxious of, or who possibly had religious objections to dogs were hypothesized to have more negative attitudes toward in-training assistance dogs in class than instructors who did not have any of these characteristics. Instructors who completed the questionnaire were separated into two groups. One group consisted of instructors who agreed or strongly agreed to either question 8 (anxious about dogs) or question 9 (allergies to dogs) in Table 1 or indicated that they were Islamic (who may or may not object to dogs as unclean). The other group consisted of all other instructors who completed the questionnaire. Consistent with the prediction, a one-tailed, two-sample, *t*-test comparing the attitudes toward in-training assistance dogs in the classroom for the first group (*M* = 2.0, *n* = 1) to the second group (*M* = 3.4, s.d. = 0.3, *n* = 7) found a statistically significant difference with a large effect size; *t*(6) = 3.846, *p* = 0.004, Cohen’s *d* = 4.112. Given that a single instructor was placed into the first group based on their responses, this result must be interpreted as exploratory and with very low confidence.

It was anticipated that the puppy raisers would report issues about bringing their in-training assistance dog to class. The mean response to questions 1 through 5 and 8 through 9 in Table 4, after reverse scoring question 8, was computed for the five puppy raisers who completed the questionnaire; *M* = 2.2, s.d. = 0.5, *n =* 5. Question 6 (professor interacts with the dog) was not included in this analysis because it may or may not be viewed as an issue. Question 7 (take dog to class when the puppy raiser has an exam) was not included because the puppy raisers typically have another person that they can leave the dog with when needed. Larger values indicate more problems that were experienced by the puppy raisers. Inconsistent with the prediction, a one-tailed, one-sample *t*-test failed to reveal a statistically significant difference above the neutral response point (3—neither agree nor disagree); *t*(4) = −3.193, *p* = 0.983, Cohen’s *d* = −1.428. Looking at the seven individual questions, puppy raisers expressed concerns only about students petting the dog without asking (*M* = 4.4, s.d. = 0.5), *W* = 15.000, *p* = 0.019, and not having enough space in the classroom for the dog to comfortably sit (*M* = 4.4, s.d. = 0.5), *W* = 15.000, *p* = 0.019.

It was predicted that the in-training assistance dogs would show few signs of stress at the point of time that data were collected, approximately 9 to 10 weeks into the semester. The mean response to questions 1 through 12 in Table 3 was computed for puppy raisers who completed the questionnaire; *M* = 1.5, s.d. = 0.3. Lower values indicate fewer signs of stress. Consistent with the prediction, a one-tailed, one-sample *t*-test found a statistically significant difference with a large effect size below the neutral response point (3—neither agree nor disagree); *t*(4) = −9.740, *p* < 0.001, Cohen’s *d* = −4.356. Looking at the 12 individual questions, the means were at or below 2.2 (approximately disagree) for each question. One puppy raiser agreed with the statement that the dog shakes and one agreed with the statement that the dog tries to get away. All other responses were at 3 (neither agree nor disagree), 2 (disagree), or 1 (strongly disagree).

Students and instructors were predicted to believe that there are ways of improving the experience of having in-training assistance dogs in class. Frequency distributions and the results of one-tailed, one-sample Wilcoxon signed-rank tests comparing the responses to each question to the neutral response point (3) are presented in Table 5.

Student scores on the Coleman Dog Attitude Scale (C-DAS) were predicted to be positively correlated with attitudes toward having in-training assistance dogs in class. Consistent with the prediction, Pearson’s *r* = 0.674, *p* < 0.001, *n* = 65.

## 4. Discussion

University students tended to have positive attitudes toward the in-training assistance dogs being in class. Although the sample size of students who reported allergies, anxiety of dogs or who were of a faith that sometimes views dogs as unclean was very limited (*n* = 3), these students did not have statistically reliably different attitudes toward the dogs than students without these characteristics. These results, based on a very small sample, suggest that bringing in-training assistance dogs to class may not be overly disruptive to the students, perhaps even to those who are otherwise sensitive to dogs. Instructors’ responses did not support positive attitudes toward these dogs but were very close to the neutral point of the Likert scale responses. This suggests that instructors have neither a positive nor negative attitude toward the in-training assistance dogs in their classes. However, a single instructor who was in the allergy, anxiety, or religious beliefs category did differ from the other instructors in their attitudes. Given the sample size, it is impossible to generalize from the single instructor to the population. Future research should strive for a larger sample of people who are allergic to dogs, anxious around dogs, or who object to dogs for religious reasons. Given the many benefits that assistance dogs provide, the results suggest that the potential costs of having in-training assistance dogs in the classroom might be outweighed by the potential benefits.

University students and instructors were offered several ideas for improving the experience of in-training assistance dogs in the classroom. While a few agreed or strongly agreed with each of the seven ideas, the only idea that received strong support was having the puppy raiser give information on how to behave around the in-training assistance dog. These results suggest that the in-training assistance dogs are not overly disruptive and that a simple change, giving information on how to behave around the dog during the first day of class, may further increase the students’ satisfaction with the dogs.

In general, most of the issues asked about the puppy raisers’ experiences were not frequently observed. Two issues, people wanting to pet the in-training assistance dog without asking and not having sufficient space in the classroom for the dog, were more frequently a problem. If the puppy raiser gave information at the start of the semester on how to behave around the in-training assistance dog, as suggested in the previous paragraph, the issue of people wanting to pet the dog without asking might be reduced. Having sufficient space in the classroom might require universities to remove or move one desk from the classroom to make extra space for the dog. This may, or may not, be an acceptable tradeoff given budgetary constraints of many universities as it might require having one less student enrolled in the class.

The puppy raisers reported few signs of stress in the in-training assistance dogs while in the classroom. This may be due in part to when the data were collected—later in the semester after the in-training assistance dogs have had a chance to be socialized and desensitized to the classroom and students. If the data were collected on the first day that the dogs visited a class, some dogs might have shown more signs of stress. Future research should consider collecting data the first time the puppy raiser brings the in-training assistance dog to class and later in the semester to see if there are differences in the signs of stress. If the in-training assistance dogs are showing signs of stress early in their socialization, it may be necessary to reduce the duration and/or frequency of classroom visits until the dog is sufficiently desensitized to the classroom environment.

This study had a limited sample size of university students who were sampled from two universities in the same state. Whether the results would generalize to other students is an open question. Given that attitudes toward the in-training assistance dogs were positively correlated with attitudes toward dogs, in general (C-DAS), and that many people tend to have positive attitudes toward dogs, it is hoped that the results will generalize.

Future research could focus on whether having the puppy raisers give information on how to behave around the in-training assistance dogs during the first day of class improves student attitudes toward having the dogs in class. While the other potential issues with the in-training assistance dogs (e.g., noise from tags, playing with toys) were not statistically significant, addressing these concerns could be tested to see if they improve attitudes in the minority of students who find these issues concerning. A larger and more diverse sample should also be tested, but ways of gaining the acceptance of student organizations of puppy raisers need to be addressed. Rather than offering a donation to the organization that provides the puppies to the student organizations, perhaps a donation directly to the student organization would increase participation. Faculty sponsors of the student organizations could also be contacted to help convince the student organizations to participate and to help gain institutional ethics approval.

Because puppies undergo cognitive, physical, and social changes as they mature, future research could address whether in-training assistance dogs have different needs at different ages. Younger puppies who have had fewer experiences with novel people and situations and who are only starting to learn the basic commands might show more signs of stress and have more classroom-inappropriate behaviors than older dogs. A longitudinal study that looks at stress, socialization, and desensitization across the first year of training would be helpful to more fully understand the needs of the in-training assistance dogs.

## 5. Conclusions

Conceptually there are many potential problems with having in-training assistance dogs in a classroom. These potential problems could impact university students, instructors, puppy raisers and the dogs themselves. This study suggests that the potential problems, at least those addressed, are not major issues for the students, instructors, puppy raisers, and the dogs. The few problems that did arise possibly could be easily addressed by having the puppy raiser make a short presentation to each class on the first day of the semester on how to behave around the in-training assistance dog. The results are encouraging as assistance dogs provide important services to many groups and university students as puppy raisers can be an important early step in the training of potential assistance dogs. The benefits of assistance animals appear to outweigh the relatively minor costs associated with having the in-training assistance dogs in the classroom.

## Figures and Tables

**Table 1 animals-15-03476-t001:** Questions about attitudes toward in-training assistance dogs in class.

I do not notice the in-training service dogs when s/he is in the classroom. Having the in-training service dog in the classroom makes the class more enjoyable. Seeing the in-training service dog in the classroom makes me happy. Having the in-training service dog in the classroom is disruptive to my teaching (instructors only)/learning (students only). Having the in-training service dog in the classroom makes me feel less stressed when going over challenging course material. I tend to focus my attention on the in-training service dog more than what I should be paying attention to in the class. I feel distracted by the in-training service dog if he/she is present on a test day. Seeing the in-training service dog makes me anxious. Having the in-training service dog in the classroom causes me to have an allergic reaction (e.g., sneezing). I feel distracted when the in-training service dog plays with a toy or moves around a lot. My mood increases after interacting with the in-training service dog if the handler allows me to pet him/her. I look forward to going to class when I know an in-training service dog will be present.

**Table 2 animals-15-03476-t002:** Questions about ways to possibly improve the experience of having an in-training assistance dog in class.

Knowing when the dog is coming to class so I can take allergy medicine. Having the dog’s handler and dog sit in the back of the classroom. Having the dog’s handler ask if it is okay for them to bring the dog to class. Being informed about how to behave around the dog before it is brought to class. Reducing noise from tags on the dog’s collar. Not allowing the dog to play with toys that make noise Having the dog only come to class on certain days depending on the activity that the class is doing.

**Table 3 animals-15-03476-t003:** Questions about signs of stress in the in-training assistance dogs.

The dog paces. The dog shakes. The dog whines. The dog barks. The dog yawns but is not tired. The dog obsessively licks something. The dog salivates more than normal. The dog’s eyes are dilated (pupils are large). The dog blinks rapidly. The dog pants even after being given water or being inside for at least 10 min. The dog tries to hide. The dog tries to get away from where you are sitting. Does the dog display these behaviors in certain classes but not others? (Yes/No) If you responded “Yes” to the previous question, please describe what, if any, any changes in the environment possibly cause this.

**Table 4 animals-15-03476-t004:** Questions about puppy raiser’s experiences of having an in-training assistance dog in class.

I feel anxious bringing my dog to class because the dog’s behavior is unpredictable. I find it distracting having the dog in class. I feel uncomfortable when people ask to pet the dog. I feel uncomfortable when people pet the dog without asking. One or more of my professors often single me out in class because I have the dog with me. One or more of my professors interact with the dog when I bring it to class. I bring the dog to class when I have a quiz or exam. My classrooms have enough space for the dog to comfortably hold a down command and be out of the way of others. I have to leave class due to the dog misbehaving.

**Table 5 animals-15-03476-t005:** Results of questions asking about ways of improving the experience of having an in-training assistance dog in class. Responses from students and faculty are combined. One person did not respond to the first item.

Question	*N* _Strongly Disagree_	*N* _Diagree_	*N* _Neither Diagree nor Agree_	*N* _Agree_	*N* _Strongly Agree_	W	*p*
Knowing when the dog is coming to class so I can take allergy medicine	26	6	35	3	3	122.0	>0.999
Having the dog’s handler and dog sit in the back of the classroom	32	16	19	6	1	207.0	>0.999
Having the dog’s handler ask if it okay for them to bring to dog to class	13	8	18	25	10	1188.5	0.799
Being informed about how to behave around the dog before it is brought to class	8	3	21	25	17	1358.0	<0.001
Reducing noise from tags on the dog’s collar	14	21	22	12	5	573.0	0.977
Not allowing the dog to play with toys that make noise	18	8	21	19	8	798.0	0.626
Having the dog only come to class on certain days depending on the activity that the class is doing	26	13	22	8	5	327.5	>0.999

## Data Availability

Data are available from the corresponding author upon request.

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
