# Peer review of "Animals2025, 15(23), 3476;https://doi.org/10.3390/ani15233476"

_animals, 2025, doi:10.3390/ani15233476_

Round 1

Reviewer 1 Report

Comments and Suggestions for Authors

This is an excellent contribution to the field.  Interesting and well written.  While the number in one group was small, it offers initial data and a great methodology for further applications and further research and deserves to be published.

It is well presented in its current format.  I only spotted 2 tiny spelling errors and I had a few other tiny points which might be worth addressing if you can easily do so

Line 67 ‘who’

It would be interesting to have some data as to how many students are puppy raisers generally and their motivations – not essential, I am just interested and think the reader would be.

Line 103 sometime’s’

Lin 168 – assistance dogs vs assistant?

You may want to explain hypothesis 3 –  it is not grounded in the literature review, it might be worth stating why the hypothesis is directional (it is not a big issue – just a little observation)

Line 471 – worth re-mentioning that it was only 3 students (I had to search back, it would be helpful to remind the reader).

final point: it is great that you are considering the dog's stress (so often this is overlooked in interventions focused on human benefit), though in an ideal world it would have included early measures of stress and then later measures (the point at which you sampled).  While I appreciate you cannot go back to do this, it would be worth mentioning in the discussion. Indeed it would be good to question the protocol and suggest that shorter introductory sessions are implemented.

All the best with publication and your future work

Author Response

Line 67 ‘who’

Thank you for catching this.  It has been corrected in the manuscript.

It would be interesting to have some data as to how many students are puppy raisers generally and their motivations – not essential, I am just interested and think the reader would be.

Thank you for your helpful suggestion.  I asked a couple of puppy raisers who are in my class this semester and both of their responses were similar – they love dogs, with the implication that raising puppies allows them to have a dog which they otherwise would not be allowed to have in student housing, and that they want to help people by working with an in-training-assistance dog.  While anecdotal, I have included these in the manuscript.

Line 103 sometime’s’

Thank you – corrected.

Lin 168 – assistance dogs vs assistant?

Thank you.  All instances of “assistant dog” have been replaced with “assistance dog”.

You may want to explain hypothesis 3 –  it is not grounded in the literature review, it might be worth stating why the hypothesis is directional (it is not a big issue – just a little observation)

Thank you for this suggestion.  We have added to the introduction that most people, including instructors and students, like dogs given the number of households in the United States that own dogs.

Line 471 – worth re-mentioning that it was only 3 students (I had to search back, it would be helpful to remind the reader).

Thank you – the sample size has been added.

final point: it is great that you are considering the dog's stress (so often this is overlooked in interventions focused on human benefit), though in an ideal world it would have included early measures of stress and then later measures (the point at which you sampled).  While I appreciate you cannot go back to do this, it would be worth mentioning in the discussion. Indeed it would be good to question the protocol and suggest that shorter introductory sessions are implemented.

Thank you for this helpful suggestion.  We have added the following to the manuscript:

Future research should consider collecting data the first time the puppy raiser brings the in-training-assistance dog to class and later in the semester to see if there are differences in the signs of stress.  If the in-training-assistance dogs are showing signs of stress early in their socialization, it may be necessary to reduce the duration and/or frequency of classroom visits until the dog is sufficiently desensitized to the classroom environment.

Reviewer 2 Report

Comments and Suggestions for Authors

Review of Animals article no 3983404

Summary

The aim of this paper is to explore understanding student, instructor and handler attitudes towards in-training assistance dogs being in college classrooms during lessons. The majority of previous research in this area has been understanding the impact of accredited assistance animals in schools and higher education settings, this study is focused dogs who have not yet entered training.

Comments regarding general concepts

This is an interesting study which looks at an area not previously explored. The references cited are appropriate and consideration should be given to additional references that support greater understanding of the needs of both the humans and animals in this context.

Articles and researchers for further consideration

  1. Brelsford, V. L., Meints, K., Gee, N. R., & Pfeffer, K. (2017). Animal-Assisted Interventions in the Classroom—A Systematic Review. International Journal of Environmental Research and Public Health14(7), 669. https://doi.org/10.3390/ijerph14070669
    • Discusses the lack of risk assessment and animal welfare
  2. Need to look at the work of Nancy Gee and Aubrey Fine
  3. Mai, J., Howell, T., Benton, P., Lewis, V., Evans, L and Bennett, P. (2021). Facilitators and Barriers to Assistance Dog Puppy Raisers’ Engagement in Recommended Raising Practices. Animals. 11. 1195. 10.3390/ani11051195.
    • Already cited by authors but also provides evidence of facilitators and barriers encountered by volunteer puppy raisers. This should be included in intro and conclusion.
  4. Morwood, S., Mai, D., Bennett, P. C., Benton, P., & Howell, T. J. (2023). Exploring the Experiences of Volunteer Assistance Dog Puppy Raisers from the Same Program at Two Australian University Campuses. Animals13(9), 1482. https://doi.org/10.3390/ani13091482
    • Similar to current study and already cited, authors should reflect how their study is different.
  5. https://iahaio.org/best-practice/white-paper-on-animal-assisted-interventions/
    • Review the IAHAIO white paper – includes definitions for AAI and considerations for human and animal wellbeing
  6. https://usaservicedogregistration.com/services/service-dog-registration/ and https://www.ada.gov/resources/service-animals-faqs/
    • Consider the information provided on these sites about the role and responsibilities of trained service dogs

Specific comments referring to line numbers, tables or figures that disclose inaccuracies within the text, or sentences that are unclear.

Line 2   Title should include the cohort of the study – e.g. college/university – classroom is not enough information

Line 9 – 23          Simple Summary could do with a restructure to start with why service animals are important and then what this study is bringing.

Line 25                Abstract could also benefit from a restructure (as above)

Line 28                Are the dogs only in classrooms or at the campus both attending classes, lectures and other spaces?

Line 29                Suggest reframe to “having an assistance dog in training in the classroom” – suggest reframe throughout to replace ‘dog/s’ with ‘assistance dog/s in training’

Line 31 – 33       Is it disruptive to the puppy raiser or distracting? Should the language used here be more training focused (strengths-based) – suggest rethinking ‘misbehaves’ to redirecting the puppy or encouraging appropriate behaviour and training (e.g. drop, stay)

Line 37 - 39        Did this study only look at students or should this be all three groups – instructors, students and volunteer puppy raisers?

Line 43 – 54       This paragraph should be restructured, it is difficult to clearly understand the why of service dogs and implications. Look to articles 1,2, 5 and 6 listed above.

Line 51 – 52       Is this the appropriate language to describe autism? Is it a disorder to be managed? Consider rewording – perhaps check how it is described in the cited paper.

Line 56 – 58       This information should be included in the abstract and simple summary

Line 56 – 65       This information needs citations – consider the articles/authors noted above.

Line 66-68         Clarify – is it the full first year of training or until the puppy reaches the age of one year?

Line 69 – 71       This sentence needs to be rewritten, e.g. “The assistance dogs in training accompany the volunteer puppy raisers to most of their daily activities, this exposes the dogs to many novel environments and people.”

Line 71 – 72       Need consistency of naming – choose between university or college students throughout the paper

Line 73 – 75       More context about the cited study would be helpful here

Line 73 – 124    This whole section would benefit from a restructure to include more concise evidence –

Line 108 – 112  This paragraph needs to be revised. Start with developmental stages of the puppies and probable ages of the puppies in training, include what actions the handlers might need to take and how that could impact all within the space (instructor, other students, handler, puppy)

Line 130 – 134  How is this related to the current study? This needs to be clarified, your study is college/university students this citation is primary aged students. Needs better linkage.

Line 136 – 143  This could be of more benefit earlier in the introduction to help the reader understand why this research is important.

Line 149 – 156  This is the first mention of costs related to puppy training – consider noting this earlier in the introduction and then clarification can be given here.

Line 158 – 162  Reconsider the language used here to describe the actions taken by the puppy raiser – is the dog misbehaving or needing instruction? Perhaps look at the guidelines from literature about the expectations of puppy raisers actions towards the puppy during the first year training and align this paragraph with those expectations. Include more citations about why the distraction of the puppy raiser or the need for care for the puppy (see IAHAIO white paper)

Line 176 – 194 Be more specific in the wording of the hypotheses – is it all students or just college/university students? H6 – how old are the puppy’s in week 9 or 10 of the semester? H7 – Are respondents changing their beliefs or increasing their understanding?

The age of the puppies should be considered throughout the study – puppy development stages may mean they are more likely to be fearful within specific ages e.g. https://reginahumanesociety.ca/programs-services/alternatives-to-admission/dog-behaviour-tips/puppy-developmental-stages-and-behaviour/ or https://www.akc.org/expert-advice/puppy-information/puppy-growth-timeline-transitions-puppyhood/

Line 207 – 209  The way the participants are described makes them identifiable – this is a concern. Is there any reason for this information to be included in this paper? Are you using this to action any statistical analyses?

Line 212 – 214 This is a more appropriate way of describing the group within the sample without being identifiable

Line 217 – 219 Is this a single donation of $100 on behalf of each institution (i.e. $200 in total) or $100 per participant?

Line 229              Why is this the first time ‘handler’ is referred to if this is the preferred term for the puppy raiser?

Line 235 -236    Good reporting of Cronbach’s alpha – gives a good indication the reader

Line 238              “..or volunteer to help children who are fearful of dogs” – how is this relevant to the current study?

Tables 1 – 3        These could be included as supplemental materials, they don’t really need to be within the text.  A single example item for each questionnaire could then be included in text to orient the reader rather than the full table.

Line 292 – 296 Where were these questions sourced from? Have they been validated as measures of stress in puppies? What about considerations for the developmental age of each puppy?

Line 342              What is IRBs?

Line 354             What considerations were given to power calculations for the results? The sample is small, are the parametric tests used appropriate for the sample size?

Line 369 – 371  This is the first mention of support for the puppy raiser (handler) – should this be included in the introduction?

Results section -            This is well written and clear.

Table 5                Consider removing the n values for each response and simplifying the table to include an overall n value or n value for each question, then the t, p and d values.

Line 480 – 481  Good reflection of the sample size limitations

Line 487 – 489 Consider citing the existing recommendations from the AKC (or similar) about providing guidelines for assistance/service dogs in public spaces

Discussion        Overall there is no reflection in the discussion on how this study is related to previous evidence or how it meets the gaps identified in the introduction – consider including citations that show how this study improved on specific understandings or reflects existing recommendations for assistance animals in public.

Line 507 – 523  The limitations and ideas for further research are well discussed here.

Author Response

We apologize in advance if we misunderstood some of your concerns which were not clear to us.  We have modified the manuscript to be consistent with our interpretation of your concerns.  If our understanding of some of your concerns is incorrect, we are happy to try to address those concerns if you will kindly provide more details about your issues.

Specific comments referring to line numbers, tables or figures that disclose inaccuracies within the text, or sentences that are unclear.

Thank you for each of the helpful suggestions given below. 

Line 2   Title should include the cohort of the study – e.g. college/university – classroom is not enough information

Corrected

Line 9 – 23          Simple Summary could do with a restructure to start with why service animals are important and then what this study is bringing.

Corrected

Line 25                Abstract could also benefit from a restructure (as above)

Corrected

Line 28                Are the dogs only in classrooms or at the campus both attending classes, lectures and other spaces?

It is up to the individual puppy raiser to decide where and when to take the in-training-assistance dog.  Because of this individual variability, this study only looks at in-training-assistance dogs in university classrooms.

Line 29                Suggest reframe to “having an assistance dog in training in the classroom” – suggest reframe throughout to replace ‘dog/s’ with ‘assistance dog/s in training’

In general, we agree with your suggestion.  In practice, it seems to be highly repetitive to use the more precise language “in-training-service dogs” in every location that the word “dog” appears in the manuscript.  As a hopefully acceptable compromise, we have added the more precise description in numerous, but not all, locations in the manuscript.  We hope that the context of a given sentence helps make it clear when we are talking about “in-training-service dogs”.

Line 31 – 33       Is it disruptive to the puppy raiser or distracting? Should the language used here be more training focused (strengths-based) – suggest rethinking ‘misbehaves’ to redirecting the puppy or encouraging appropriate behaviour and training (e.g. drop, stay)

We have changed the language to clarify that managing the dog’s behavior is disruptive to the puppy raiser if they must direct their attention away from classroom activities to the in-training-assistance dog.

Line 37 - 39        Did this study only look at students or should this be all three groups – instructors, students and volunteer puppy raisers?

Thank you.  We have clarified that it is all three groups.

Line 43 – 54       This paragraph should be restructured, it is difficult to clearly understand the why of service dogs and implications. Look to articles 1,2, 5 and 6 listed above.

We are sorry, but we do not understand what you mean by “the why of service dogs”. Our interpretation of your concern must be incorrect as we believe that the explanation of why assistance dogs exist and the functions that they serve are clearly delineated in this paragraph.

Line 51 – 52       Is this the appropriate language to describe autism? Is it a disorder to be managed? Consider rewording – perhaps check how it is described in the cited paper.

Thank you.  We have updated this sentence with language from one of the cited articles.

Line 56 – 58       This information should be included in the abstract and simple summary

We apologize in advance if we are misunderstanding what you are asking for.  In the manuscript, lines 56-58 state:

Before dogs can be certified as assistant dogs, they undergo two types of training: general training (socialization, desensitization, and obedience training) and specialized training for the group that they will eventually serve.

Both the simple summary and abstract address the general training which is what is addressed in the manuscript.  Because the manuscript is not about the specific training that the dogs undergo, we are not sure how including such information is relevant to the abstract and simple summary.

Line 56 – 65       This information needs citations – consider the articles/authors noted above.

While some of the references (1, 2, 5, and 6) you cited above would be useful if the current research was on animal-assisted interventions, the in-training-assistance dogs we studied are fundamentally different than the dogs typically used in AAI – they are puppies rather than adults, have received, in some cases, far less training than dogs used in AAI, and will ultimately serve very different functions.  That makes it difficult to generalize studies on and guidelines for AAI dogs to in-training-assistance dogs.  To include such recommendation would be misleading at best. 

Line 66-68         Clarify – is it the full first year of training or until the puppy reaches the age of one year?

Thank you.  We have changed “life” to “training”.

Line 69 – 71       This sentence needs to be rewritten, e.g. “The assistance dogs in training accompany the volunteer puppy raisers to most of their daily activities, this exposes the dogs to many novel environments and people.”

We have modified your recommended sentence and inserted it into the manuscript.

Line 71 – 72       Need consistency of naming – choose between university or college students throughout the paper

Thank you.  All occurrences of “college” have been replaced by “university”.

Line 73 – 75       More context about the cited study would be helpful here

The sentence at line 73 is a sentence that introduces the general topic to be addressed in the paragraph and the next paragraphs.  Supporting citations are given for each specific issue addressed in these paragraphs.  We have added “In a questionnaire study of adults living in Adelaide Australia, …”

Line 73 – 124    This whole section would benefit from a restructure to include more concise evidence –

We respectfully disagree.  This section addresses the many reasons why there might be issues associated with bringing in-training-assistance dogs into a classroom.  It establishes the background for one of the main topics of the research.  Fifty lines of text, approximately one page, is not unreasonable for the background of one of the main topics of research.

Line 108 – 112  This paragraph needs to be revised. Start with developmental stages of the puppies and probable ages of the puppies in training, include what actions the handlers might need to take and how that could impact all within the space (instructor, other students, handler, puppy)

While it would be interested to do this, this study did not address developmental issues associated with the in-training-assistance dogs.  While it would be relevant for future research that addressed this issue, we feel that it is irrelevant to the purpose of the current study.  We have included this topic as potential future research in the discussion section.

Line 130 – 134  How is this related to the current study? This needs to be clarified, your study is college/university students this citation is primary aged students. Needs better linkage.

We have added the following sentence to help establish the linkage:

While the preceding studies look as younger students, it may be that these benefits also apply to older students. 

Line 136 – 143  This could be of more benefit earlier in the introduction to help the reader understand why this research is important.

The fact that not all in-training-assistance dogs will become service dogs is a minor point in why the research is important.  The importance of the research, as described earlier in the introduction, is that having any dog in the classroom might be problematic for the students and instructors for the several reasons given.

Line 149 – 156  This is the first mention of costs related to puppy training – consider noting this earlier in the introduction and then clarification can be given here.

We again apologize if we are misunderstanding your concern.  The costs (as in cost-benefit analysis as given by “to compare the costs to the benefits”) related to puppy raising are discussed starting with the paragraph “Bringing in-training-assistance dogs to class…” and continue for the next eight paragraphs.

Line 158 – 162  Reconsider the language used here to describe the actions taken by the puppy raiser – is the dog misbehaving or needing instruction? Perhaps look at the guidelines from literature about the expectations of puppy raisers actions towards the puppy during the first year training and align this paragraph with those expectations. Include more citations about why the distraction of the puppy raiser or the need for care for the puppy (see IAHAIO white paper)

We have adjusted the language to “If the puppy raiser must stop paying attention to class and manage the in-training-assistance dog’s behavior, the puppy raiser might miss important information that is being presented by the instructor of the class.”

The IAHAIO white paper gives best practices for animal assisted interventions.  The in-training-assistance dogs are not providing animal assisted intervention.  Because animals in AAI are typically older and have received more training than in-training-assistance dogs, it is difficult to generalize from AAI to in-training-assistance dogs.  Thus, we believe that the IAHAIO recommendations are not relevant to the current research and that it would be misleading to suggest that the in-training-assistance dogs should be held to the same standard as AAI dogs.

Line 176 – 194 Be more specific in the wording of the hypotheses – is it all students or just college/university students? H6 – how old are the puppy’s in week 9 or 10 of the semester? H7 – Are respondents changing their beliefs or increasing their understanding?

We have added “university” to “students”.  We did not collect data on the in-training-assistance dog’s ages, so we are unable to provide the requested information.  We did not provide to our sample information on why in-training-assistance dogs in the classroom might be beneficial.  Thus, we did not try to manipulate their understanding but rather measured their attitudes about this topic.

The age of the puppies should be considered throughout the study – puppy development stages may mean they are more likely to be fearful within specific ages e.g. https://reginahumanesociety.ca/programs-services/alternatives-to-admission/dog-behaviour-tips/puppy-developmental-stages-and-behaviour/ or https://www.akc.org/expert-advice/puppy-information/puppy-growth-timeline-transitions-puppyhood/

Thank you for this suggestion.  We did not record the age of the in-training-assistance dogs, so we are unable to look at age differences.  We have included your helpful suggestion in the discussion section as possible future research.

Line 207 – 209  The way the participants are described makes them identifiable – this is a concern. Is there any reason for this information to be included in this paper? Are you using this to action any statistical analyses?

We are sorry, but we do not understand your concern.  The researchers and readers do not have access to class rosters, and the instructors do not have access to the data.  Thus, there is no way for either the instructors or researchers to link data to individuals.  Therefore, the individual’s responses remain anonymous. The results are reported in aggregate and not as individual responses. Yes, as we are sure that you are already aware, there is a reason for including basic demographic information in the manuscript – it helps the reader to decide how well the data can generalize from the sample to a given population.

Line 212 – 214 This is a more appropriate way of describing the group within the sample without being identifiable

Please see our reply to your previous concern.

Line 217 – 219 Is this a single donation of $100 on behalf of each institution (i.e. $200 in total) or $100 per participant?

As stated in the manuscript, “a single $100 donation would be made” and not “a single $100 donation would be made for each institution” or “a single $100 donation would be made per participant.”

Line 229              Why is this the first time ‘handler’ is referred to if this is the preferred term for the puppy raiser?

“Puppy raiser” is the term used in the literature.  “Handler” is the term used by the organization that provides the dogs and is the term that the puppy raiser/handlers use when talking about their role in raising the dogs to their classmates.  Thus, the sample is likely more familiar with the term “handler” than “puppy raiser”.  As stated in the manuscript, “the term handler was used throughout the questionnaire as this is the common term in the sample for puppy raiser.”

Line 235 -236    Good reporting of Cronbach’s alpha – gives a good indication the reader

Line 238              “..or volunteer to help children who are fearful of dogs” – how is this relevant to the current study?

These are reported to establish the validity of the C-DAS as stated in the next sentence in the manuscript.  We have more explicitly stated this in the revised manuscript.

Tables 1 – 3        These could be included as supplemental materials, they don’t really need to be within the text.  A single example item for each questionnaire could then be included in text to orient the reader rather than the full table.

While it is often the case that the questionnaires are included as supplemental materials, the material and results sections refers to individual questions when mentioning how the questionnaires were scored.  To separate the questions from the manuscript would decrease the clarity of the scoring and force the reader to switch between documents to understand a basic methodological issue.

Line 292 – 296 Where were these questions sourced from? Have they been validated as measures of stress in puppies? What about considerations for the developmental age of each puppy?

The questions were created by the authors based on signs of stress given in the citation given in the first sentence of the paragraph.

Line 342              What is IRBs?

Corrected with “Institutional Review Boards (ethics committees)”.

Line 354             What considerations were given to power calculations for the results? The sample is small, are the parametric tests used appropriate for the sample size?

We sampled as many students, instructors, and puppy raisers as we could given our time constraints – the first author graduated in May of 2025.  We agree that larger sample sizes would be better, both from a statistical power perspective and from a generalizability perspective, but it was not possible.  We acknowledge the limitations of the sample size in both the results and discussion sections where appropriate.

Line 369 – 371  This is the first mention of support for the puppy raiser (handler) – should this be included in the introduction?

We have added the following sentence to the introduction

Some of these issues may be at least partially mitigated by leaving the in-training-assistance dog with another puppy raiser when the need arises.

Results section -            This is well written and clear.

Table 5                Consider removing the n values for each response and simplifying the table to include an overall n value or n value for each question, then the t, p and d values.

We believe that showing the distribution of responses and not just sample size includes important information about the results.  For example, giving the sample size for the first question (n = 73) does not convey that the majority of responses were strongly disagree or neutral and that only a small number (6) of the responses are agree or strongly agree with the statement. That is, the distribution of responses are important descriptive statistics.

Line 480 – 481  Good reflection of the sample size limitations

Line 487 – 489 Consider citing the existing recommendations from the AKC (or similar) about providing guidelines for assistance/service dogs in public spaces

We apologize in advance if we are not correctly understanding your concern.  The manuscript is not about assistance/service dogs.  Including guidelines for assistance/service dogs in public spaces is, in our opinion, not relevant to the topic of the manuscript.  We believe that it is unreasonable to expect in-training-assistance dogs, who are puppies and may not be fully trained, to be expected to behave in the same way that an adult, extensively trained assistance dog should behave.

Discussion        Overall there is no reflection in the discussion on how this study is related to previous evidence or how it meets the gaps identified in the introduction – consider including citations that show how this study improved on specific understandings or reflects existing recommendations for assistance animals in public.

We again apologize that we must be misunderstanding your concerns.  The reason that there is no reflection comparing the current study to previous literature is that there is no previous literature on having in-training-assistance dogs in the classroom.  Comparing in-training-assistance dogs to dogs used in AAI or to assistance dogs that have completed their training is like comparing apples to oranges – puppies that in some cases have received very little training in the basic commands are not comparable to adult dogs who have received, in some cases, very extensive training.  This makes generalizing from one set of dogs to the other questionable at best.

The gaps in the literature – do in-training-assistance dogs in the classroom present problems to university students, instructors, and handlers, and whether the in-training-assistance dogs show signs of stress – are addressed in the discussion.

The study is not about assistance animals in public and can make no recommendation for them.  We do make recommendations for in-training-assistance dogs in classrooms that are warranted by our results.

Line 507 – 523  The limitations and ideas for further research are well discussed here.

Reviewer 3 Report

Comments and Suggestions for Authors

Thank you for highlighting this area and for suggesting some ways that could improve the situation for students who are puppy raising. Please take the following comments in the spirit that they were intended, namely to assist making your article the best it can be.

Major points:

The main point I believe needs clarification is your use of the terms "socializing" vs. "desensitizing". Socialization involves the exposure of dogs, usually young pups, to a variety of people, animals, environments and objects. It is not just people. Desensitization involves reducing the reaction of the animal to a stimulus, usually by slowly increasing the level of that stimulus, e.g. slowly increasing the volume of a sound that produces a fear response, staying at a level where no fear is produced. There are several places in the Introduction & Discussion where the use of these terms needs clarification. I do understand that there is some confusion over the terms by some authors. 

The next area is your inconsistent use of assistant dog & assistance dogs. In my experience, the term "assistance dogs", or "service dogs in the US, is the term used and I have not come across using "assistant dogs". Lines 43, 56, 66, 73, 113, 115

Finally, can you justify using parametric tests such as the one-sided t-test when dealing with Likert scales? Was there a normal distribution? Did you consider using nonparametric tests?

Minor points:

Line 33 - "misbehaves" is a subjective term. To manage the dog's behavior

would be more accurate.  

Line 67 - consider adding 'who' - i.e. puppy raisers, who assist

Line 69 - "members of organizations" is very vague. Give an example such as the cited reference, e.g. high school students in FFA programs

Lines 85-90. This is an extremely long sentence. Consider rephrasing to make it clearer. 

Line 159. Use of "misbehaves" is subjective. 

Line 358. Please define the neutral point, i.e. "neither agree nor disagree". This is not done until much later in your article.

Line 410. n = ? Participants section states that there were 9 instructors but testing of H4 compares n=1 to n=7 (total of n=8) so how many participated in testing H3?

Line 446. n=? Did all 5 handlers complete this section?

Line 467. n=?

I agree that you could have increased participation rates by providing payment to the student organization rather than the dog organization. As your numbers are low, especially for instructors, this paper will be of interest but not really open to general interpretation. Best of luck. 

Author Response

The main point I believe needs clarification is your use of the terms "socializing" vs. "desensitizing". Socialization involves the exposure of dogs, usually young pups, to a variety of people, animals, environments and objects. It is not just people. Desensitization involves reducing the reaction of the animal to a stimulus, usually by slowly increasing the level of that stimulus, e.g. slowly increasing the volume of a sound that produces a fear response, staying at a level where no fear is produced. There are several places in the Introduction & Discussion where the use of these terms needs clarification. I do understand that there is some confusion over the terms by some authors. 

Thank you for clarifying the distinction between socialization and desensitization.  While the desensitization that you mention is one way of reducing an animal’s response to stimuli, there are other ways to desensitize an animal, such as repeatedly presenting an animal with a stimulus or situation without harm.  The animal learns that it won’t be harmed by that stimulus/situation and its response is often reduced.  That is what happens with these in-training-assistance dogs.

Hopefully, our rewording of socialization and desensitization in the manuscript are clarified.

The next area is your inconsistent use of assistant dog & assistance dogs. In my experience, the term "assistance dogs", or "service dogs in the US, is the term used and I have not come across using "assistant dogs". Lines 43, 56, 66, 73, 113, 115

Thank you.  All instances of “assistant dog” have been replaced with “assistance dog” in the manuscript.

Finally, can you justify using parametric tests such as the one-sided t-test when dealing with Likert scales? Was there a normal distribution? Did you consider using nonparametric tests?

While the response distribution to individual questions may not be normally distributed, when multiple questions are combined to form a single measure, the measure will be approximately normal according to the central limit theorem.  The t-test is robust to violations of the assumption of normality when the sample size is sufficiently large.  That is likely the case for the student data, but not for the instructor and puppy raiser data.

We agree that for the cases where the sample size is small and/or we are looking at individual questions, it may is more statistically appropriate to use non-parametric statistics.  We have replaced the parametric tests with their non-parametric counterparts where appropriate.

We have added the following sentences to section 2.4 Statistical Analysis:

Even though the responses to the individual questions may not have been normally distributed, once the responses are summed across questions, the measure will be approximately normal by the central limit theorem. 

Minor points:

Thank you for pointing these out.  Corrections for each minor point have been made in the manuscript.

Line 33 - "misbehaves" is a subjective term. To manage the dog's behavior

would be more accurate.  

Thank you for this suggestion.  We have changed this and the occurrence at line 159 to “manage the dog’s behavior”.

Line 67 - consider adding 'who' - i.e. puppy raisers, who assist

Corrected

Line 69 - "members of organizations" is very vague. Give an example such as the cited reference, e.g. high school students in FFA programs

Additional information has been added

Lines 85-90. This is an extremely long sentence. Consider rephrasing to make it clearer. 

We have separated the material into four sentences.

Line 159. Use of "misbehaves" is subjective. 

Corrected

Line 358. Please define the neutral point, i.e. "neither agree nor disagree". This is not done until much later in your article.

We have added this information to the manuscript at the indicated location.

Line 410. n = ? Participants section states that there were 9 instructors but testing of H4 compares n=1 to n=7 (total of n=8) so how many participated in testing H3?

Thank you.  We have clarified that there were only eight instructors who completed this questionnaire.

Line 446. n=? Did all 5 handlers complete this section?

Thank you.  We have added that five handlers completed this questionnaire.

Line 467. n=?

Thank you.  We have clarified that n = 65 for the correlation.

I agree that you could have increased participation rates by providing payment to the student organization rather than the dog organization. As your numbers are low, especially for instructors, this paper will be of interest but not really open to general interpretation. Best of luck. 

Round 2

Reviewer 2 Report

Comments and Suggestions for Authors

The updated manuscript has adequately addressed the original review and the author's notes have clarified all comments.